# The Association between Lifestyles (Walking/Diet) and Cultural Intelligence: A New Attempt to Apply Health Science to Cross-Cultural Research

**DOI:** 10.3390/bs14010028

**Published:** 2023-12-29

**Authors:** Keisuke Kokubun, Kiyotaka Nemoto, Yoshinori Yamakawa

**Affiliations:** 1Open Innovation Institute, Kyoto University, Kyoto 606-8501, Japan; yamakawa@bi-lab.org; 2Graduate School of Management, Kyoto University, Kyoto 606-8501, Japan; 3Department of Psychiatry, Institute of Medicine, University of Tsukuba, Tsukuba 305-8577, Japan; kiyotaka@nemotos.net; 4Institute of Innovative Research, Tokyo Institute of Technology, Tokyo 152-8550, Japan; 5ImPACT Program of Council for Science, Technology and Innovation, Cabinet Office, Government of Japan, Tokyo 100-8914, Japan; 6Office for Academic and Industrial Innovation, Kobe University, Kobe 657-8501, Japan; 7Brain Impact, Kyoto 606-8501, Japan

**Keywords:** cultural intelligence, curiosity, grit, dietary balance, walking, health science

## Abstract

A growing amount of research is being conducted on cultural intelligence (CQ), which is the ability to adjust and adapt successfully to a variety of surroundings. CQ is a vital quality for people in diversified societies, as are seen today. However, it is still unclear how an individual can develop or strengthen CQ because previous studies have shown that variables such as foreign experience and personality are not exclusively sufficient as CQ antecedents. If CQ can be treated health-scientifically, as the CQ developers argue, diet and exercise that are effective in keeping the body and brain healthy may also correlate with CQ. It is of great significance to clarify the relationship between lifestyle and CQ by applying recent previous research showing the relationship between CQ and both the brain and intelligence, as well as between lifestyle and both the brain and intelligence. Using data derived from 142 Japanese businesspersons, the results of structural equation modeling indicate that lifestyles (dietary balance and walking frequency) are significantly associated with CQ after controlling for personalities (curiosity and grit) and international experiences (length of studying and working abroad), which have been used as predictors of CQ in previous studies. Furthermore, the moderation test showed that the effect of dietary balance on increasing CQ is greater for people with less overseas experience, indicating that dietary balance compensates for lack of overseas experience in the effect of maintaining the level of CQ. These suggest the effectiveness of a health-scientific approach to the influencing factors of CQ. This study is the first to show that CQ is influenced by lifestyle such as walking and dieting, in addition to personality and overseas experience, and will contribute to the future development of health science and cross-cultural research.

## 1. Introduction

A growing amount of research focuses on cultural intelligence (CQ), which is the ability to adapt successfully to different cultural contexts. This is a vital quality for people in diversified societies that are seen nowadays [1]. CQ comprises the metacognitive, cognitive, motivational, and behavioral dimensions, each containing various essences that are necessary for cross-cultural adaptation [2]. Metacognitive CQ reflects mental processes that individuals use to acquire and understand cultural knowledge; cognitive CQ reflects knowledge of the norms, practices, and conventions in different cultures acquired through education and personal experiences; motivational CQ reflects the capability to direct attention and energy toward learning about and functioning in situations characterized by cultural differences; and behavioral CQ reflects the capability to exhibit appropriate verbal and nonverbal actions when interacting with people from different cultures [3]. CQ has been linked to various skills and behaviors, including cultural adaptation and performance [4], organizational citizenship behavior [5], and work engagement [6].

However, as compared to the outcomes of CQ, the research on the antecedents of CQ is still inadequate [7]. Therefore, it remains rather unclear how an individual develops or strengthens CQ. The majority of articles regarding the antecedents focus on the international experience or cultural exposure (including related training/education) to develop CQ [8,9,10], while others examine individual differences in personality traits [11,12,13]. However, the relationship between both overseas experience and personality traits and CQ is not strong [3,7]. Therefore, a significant factor that could be improved upon is the comprehensiveness of antecedent variables of CQ, which has not been seen in previous studies. In this sense, we have not come across any study that has employed variables related to lifestyles. If CQ is related to brain activities as advocated by CQ developers [14,15], it is worthwhile to test the association between lifestyle variables and CQ. This is because research in brain science has clarified the relationship between lifestyles and brain conditions [16,17,18], which indicates the consequent association between lifestyles and CQ. Based on this understanding, in this research, we test the association of CQ with such lifestyle variables, adding to the variables related to international experience and personality traits by using businesspersons who were recruited in Japan. As CQ scholars have called for a greater examination of CQ antecedents beyond international experience and personality [7,19], we assume that the current study could respond to such calls. This paper is the first study to examine the possibility that lifestyle variables can be predictors of CQ from a health science perspective. To this end, we use structural equation modeling (SEM) to show that two lifestyle variables (walking frequency and dietary balance) are correlated with CQ when controlling for personalities (curiosity and grit) and international experiences (length of studying/working abroad), which have been used as predictor variables of CQ,

## 2. Literature Review

A recent study indicates that two personality traits, which are curiosity (or exploration and perseverance) and immersion experiences (length of stay in foreign countries), are significantly correlated to the level of CQ [20]. This is consistent with past studies that have found that the length of stay [10,21] and tolerance and curiosity lead to higher CQ [22,23].

However, a comprehensive review indicates that the relationship between international experience and CQ is not strong enough to establish a significant link between the variables [7]. Also, personalities, including curiosity and grit (or perseverance, tolerance, etc.), may not be considered as exclusive determinants. CQ is conceptualized as a state-like construct that consists of malleable abilities that can be developed, while personality traits are generally stable and are not dependent on the situation [3]. These observations and discussion indicate that antecedents other than international experiences and personalities need to be studied to further understand CQ.

A hint when considering other factors that can influence CQ is the relationship between CQ and neuroscience proposed by the CQ developers [14,15]. Organizing a series of previous studies, they proposed cortical regions as neurological mediators of cognitive (i.e., medial–temporal/diencephalic and neocortical regions), metacognitive (i.e., anterior rostral media frontal cortex, including the paracingulate cortex), motivational (i.e., orbitofrontal cortex), and behavioral CQ (i.e., posterior rostral medial frontal cortex and dorsal anterior cingulate cortex). As far as the researchers know, no research directly verifies these proposals. However, empirical studies in cultural neuroscience have shown supportive results. For example, a brain imaging study conducted by Hedden et al. [24] showed that, when engaged in a task that was not congruent with their cultural values, people required increased attention, and their frontal and parietal regions that control attention were activated to a greater extent (see also the narrative review by Chang [25]). This suggests that keeping the brain healthy and active is important for maintaining high levels of CQ.

Possible factors that have an impact on the body and brain are quite diverse, and these also include food and exercise. For example, a previous study has shown that dietary balance correlates with brain health (measured by gray matter volume) in healthy young and middle-aged adults [26]. Also, a previous study of college male students has shown that walking and running increase the Brain-Derived Neurotrophic Factor (BDNF), the nervous system humoral protein essential for the growth of brain cells [27]. Furthermore, emerging research indicates that the effect of diet on the brain is integrated with exercise [28,29] because the combination of certain diets and exercises can affect mitochondrial energy production and can have additive effects on synaptic plasticity and brain function [28]. Moreover, physical exercise facilitates the release of neurotrophic responses such as peripheral brain-derived neurotrophic factors [30], increases blood flow, improves cerebrovascular health, and benefits glucose and lipid metabolism, carrying food to the brain [31]. These indicate that we can consider lifestyle, including food and exercise, as a possible antecedent of CQ if we regard the latter as derived from brain activities.

Consistent with these results from brain science research, traditional health science research has shown that diet and exercise are predictors of cognitive ability [32,33]. In addition, recent research has shown that diet and exercise may be predictors of emotional intelligence (EI)—the ability to recognize, control, and evaluate emotions [34]. However, to the authors’ knowledge, there are no studies that have shown a relationship between either diet or exercise and CQ. As these positive correlations between intelligence and CQ have been confirmed in previous studies [35], it may be meaningful to test lifestyle variables as predictors of CQ, as has been performed for other bits of intelligence. Sleep is another typical predictor of the brain and intelligence, but it has been pointed out that it has received less attention from researchers than diet and exercise [36]. Therefore, we test the association of CQ with the most representative lifestyle variables (walking and diet) while using Japanese businesspersons as the subject. The reason why we chose Japanese people as the research target is that ethnocentrism characteristics in Japanese companies have often been pointed out for making it difficult for them to collaborate with foreigners in a cross-cultural environment due to the homogeneous culture to which they are accustomed [37,38]. Nevertheless, as far as the authors know, no efforts have been made to elucidate the antecedents of Japanese CQ.

Figure 1 is the analytical model of this study. We clarify the relationship between the two variables representing lifestyle (walking frequency and dietary balance) and the CQ, consisting of four subsets shown in yellow. To this end, we use two variables representing personality (curiosity and grit) and two variables representing international experience (length of studying/working abroad), which have been established to have a relationship with CQ in previous research, as covariates and moderation variables.

## 3. Theoretical Framework

Previous studies have shown that the state of the brain in a healthy person, as measured by gray matter volume, is associated with dietary balance [26]. Likewise, walking and running have been shown to increase the Brain-Derived Neurotrophic Factor (BDNF) [27]. Emerging research also indicates that the effect of diet on the brain is integrated with exercise [28,29,30,31]. Consistent with these results in the brain science field, in the health science field, the relationship between diet and exercise and intelligence such as cognitive function and EI is being clarified [32,33,34]. Since CQ has been confirmed to be related to the brain and intelligence [24,25,35], it is possible that CQ has these lifestyle variables as predictors, like the brain and intelligence. Therefore, the following hypotheses are proposed:

**H1.** 
*Dietary balance is positively related to CQ.*


**H2.** 
*Walking frequency is positively related to CQ.*


## 4. Research Methodology

### 4.1. Participants

A total of 142 participants (30 females and 112 males) working in 27 companies were recruited in Tokyo, Japan. They were all Japanese businesspersons living in Japan, except for three Japanese persons working in affiliates in Shang Hai, China. (These three people were included in the sample because they lived in large cities where it was easy to reproduce the Japanese lifestyle, such as where Japanese food was easily available and where living areas were well-developed and walkable.) Also, the participants were all knowledge workers who attended a study session on brain activities held by the authors. However, there were no criteria for participation related to CQ, such as whether or not they had overseas experience or how many efforts were being made to internationalize the workplace. The questionnaire survey was conducted at the Tokyo Institute of Technology from September to December 2019. Participants visited the designated venues one by one and answered the online questionnaire on our personal computers. According to the self-report, no subjects recruited had records of neurological, psychiatric, or other medical conditions that could affect the central nervous system. This study was approved by the Ethics Committee of Kyoto University (Approval Number 27-P-13) and Tokyo Institute of Technology (Approval Number A16038) and was conducted following the institutes’ guidelines and regulations. All participants provided written informed consent before participation, and their anonymity was maintained.

### 4.2. Measures

We used different sources for measuring each factor.

#### 4.2.1. Curiosity

Curiosity is the motivational state that drives us toward the information expected to fill the specific gap in one’s knowledge [39]. It has been identified that people must have cultural curiosity and should be tolerant of uncertainty and ambiguity to be able to successfully deal with cultural obstacles in multicultural settings [40]. Therefore, empirical research indicates that curiosity is connected to the motivation to acquire new knowledge and experience, and facilitates exploratory behavior [41,42] and cross-cultural adjustment [43]. Curiosity was measured by a Japanese version [44] of the 5-point Likert scale, the Trait Curiosity and Exploration Inventory-II [45] scale, which contains 10 items, including “I actively seek as much information as I can in new situations.”

#### 4.2.2. Grit

Grit is defined as trait-level perseverance and the passion required to achieve long-term goals [46]. Grit was measured by a Japanese version [47] of the 5-point Likert scale, the Short Grit Scale, developed by Duckworth and Quinn [48]. This contains eight items, including “I finish whatever I begin”. It has been observed that international students who have a more tolerant attitude have fewer problems with intercultural adaptation, as interacting with people from other cultures can be stressful, challenging, and tiring [49].

Previous studies indicate that two personality traits, which are curiosity (which is measured by the Trait Curiosity and Exploration Inventory-II [45] scale) or perseverance (which is a subscale of grit and can be measured by the Short Grit Scale [48] and length of stay in foreign countries, are significantly related to the level of CQ [20]. Likewise, tolerance and curiosity, which are amongst the six sub-facets of openness to experience (which are intellectual efficiency, ingenuity, curiosity, aesthetics, tolerance, and depth) have the strongest impact on CQ [22,23].

#### 4.2.3. Dietary Balance

Dietary balance was determined using an abridged eleven-item Food Diversity Score Kyoto (FDSK-11) [50]. The FDSK-11 consists of eleven main food groups (grain, meat, fish and shellfish, eggs, milk, beans and soybean products, potatoes, vegetables, seaweed, nuts, and fruits). Participants were asked whether they had eaten items that belonged to each of the eleven food groups for a day or more (a score of 1) or a week or less (a score of 0). The scores of 11 questions were aggregated to provide an FDSK-11 ranging from 0 to 11, with a higher score indicating greater dietary balance. In previous research, FDSK-11 was closely associated with activities of daily living, depression, and subjective quality of life [50,51].

#### 4.2.4. Walking Frequency

The walking frequency was measured using the question, “During the last week, how many days did you walk for at least ten minutes at a time? This includes walking at work and home, walking to travel from one place to another, and any other walking that you could have done solely for recreation, sport, exercise, or leisure”. This was adapted from the Japanese version [52] of the International Physical Activity Questionnaire (IPAQ)—Short Form [53]. IPAQ was used because it is an international standard index that has been translated into the languages of many countries including Japan and used for diverse studies. For instance, previous research indicates that the walking measured by IPAQ was correlated with the total number of accelerometer-based steps of healthy young adults [54].

#### 4.2.5. International Experiences

The relationship between international experience and CQ has been well-established [55,56]. Previous studies indicate that the “length” of stay in foreign countries leads to a higher CQ [10,20,21,57,58] and that previous international work [57,59] and nonwork [10,21] experiences are positively related to CQ. However, recent studies indicate that education or working abroad leads to a higher CQ as compared to other experiences such as international vacations [9,60]. Therefore, we employ the phrase “length of studying abroad” (instead of a phrase like “non-work international experience”) to exclude international experiences based on other purposes (for example, vacations), with the phrase “length of working abroad” as the explanatory variables, with the expectation that the subjects’ previous international work and study experiences will have a positive influence on their CQ. Length of working abroad was measured through a new question, “For how many months have you lived in foreign countries for work?” Length of studying abroad was also measured through a new question, “For how many months have you lived in foreign countries for study?” We use these variables as control variables in the following analysis.

#### 4.2.6. CQ

As globalization progresses, indicators that objectively show people’s ability to adapt to different cultures are being actively developed. This study uses CQ, the Cultural Intelligence [3] scale, as the dependent variable because CQ is one of the most actively cited indicators in multicultural settings and, therefore, has been translated into many languages, including Japanese. Furthermore, interestingly, the indicator’s developers have discussed its relationship to brain science [14,15]. In the current research, CQ was measured using a Japanese version [61] of the 7-point Likert scale, the Cultural Intelligence Scale, developed by Ang et al. [3]. CQ is a multidimensional construct with twenty items that make up the four sub-dimensions of metacognitive (four items, including “I am conscious of the cultural knowledge I use when interacting with people with different cultural backgrounds”), cognitive (six items, including “I know the legal and economic systems of other cultures”), motivational (five items, including “I enjoy interacting with people from different cultures”), and behavioral (five items, including “I change my verbal behavior (e.g., accent, tone) when a cross-cultural interaction requires it”) CQ. Each dimension represents distinctive abilities that are applicable in cross-cultural environments and can be aggregated to an overall CQ [3].

## 5. Analysis

All statistical analyses have been performed using IBM SPSS Statistics/AMOS Version 26 (IBM Corp., Armonk, NY, USA). Before proceeding to the main analyses, Harman’s single-factor analysis was used to check if the variance in the data could be largely attributed to a single factor, while the confirmatory factor analysis (CFA) was used to test if the factors were related to the measures. First, the factor analysis indicated that only 32.2 percent of the variance could be explained by a single factor, which was <50 percent. Thus, it was established that the data did not suffer from the common method variance [62]. Next, for CFA, the model fit was evaluated by examining the indices recommended by Hu and Bentler [63]. These were the ratio of Chi-Square to the degree of freedom (χ^2^:df ratio: acceptable if 2.0 or less; Bollen [64]), the comparative fit index (CFI: good if 0.90 or more; Bentler [65]), the root-mean-square error approximation (RMSEA; good if 0.06 or less), and the standardized root-mean-square residual (SRMR; good if 0.08 or less; Bentler [65]). To begin, a six-factor model was tested with metacognitive CQ, cognitive CQ, motivational CQ, behavioral CQ, curiosity, and grit, each loading on a single factor. The results indicated a good fit between the data and the model. Next, a three-factor model was tested, in which the subscales of CQ were combined, and curiosity and grit were treated as single factors. The results also yielded a good fit. Lastly, a one-factor model was tested, where all the factors were combined into one single factor. These results indicated a bad fit in terms of RMSEA and SRMR. Results of the same are presented in Table 1. Therefore, in the following analysis, CQ is used as a latent variable and CQ subsets are used as manifest measures to clarify the factors that influence CQ. We treat all antecedent variables and the four CQ subsets as manifest indicators of the structural models. In other words, we treat all the CQ subsets as parceling items by first-order constructs [66].

However, Podsakoff et al. [67] identified several limitations of the above common method factor technique and concluded that procedural remedies, such as examining interaction effects, are the strongest options to rule out common method bias. Moreover, simulation studies [68] and mathematical proof [69] indicated that common method effects do not impact interaction effects. Therefore, we examine interactive effects between variables to further address common method concerns in our data and strengthen the theoretical contribution in this field.

## 6. Results

Descriptive statistics of the subjects and correlation coefficients between the psychological scales are shown in Table 2. Neither gender nor age showed a correlation with the four subsets of CQ at the 5% level. This indicates that the analysis results shown below are not due to the attributes of the subject. On the other hand, the four main variables, from grit to walking frequency, and two international experience variables show a significant correlation at the 5% level with most of the four subsets of CQ. The results of the structural equation modeling are summarized in Figure 2. Personality traits (curiosity and grit) and lifestyles (dietary balance and walking frequency) are positively associated with CQ, thus supporting H1-4. The length of working abroad is associated with a CQ subset, cognitive CQ. On the other hand, curiosity was correlated with motivational CQ in addition to CQ. Dietary balance was correlated with metacognitive CQ in addition to CQ.

Standard deviations of the length of working or studying abroad were significantly larger than the mean, indicating that some participants studied or worked abroad for considerable periods, while others did not have this experience at all. In these cases, it must be determined if there was a difference between those with no experience and any experience in terms of the levels of CQ and its subsets. Therefore, we alternatively employed dichotomous values (international experience = 1; no international experience = 0) for the length of working or studying abroad as the independent variables. However, none of them show significant differences, indicating that short international experiences have a similar impact on CQ as longer international experiences. Furthermore, even when data from the three people living in China were excluded, there was no significant difference in the analysis results (omitted in the figure but available upon request).

Furthermore, the mean-centered interaction terms of the four main explanatory variables and the two foreign experience variables were input. As a result, only the interaction term of length of studying abroad and dietary balance became significant. To further understand the meaning of the significant interaction terms, in Figure 3, the data were divided into a group with high length of studying abroad and a group with low length of studying abroad, in which the horizontal axis shows the group with high dietary balance and the group with low dietary balance and the vertical axis shows CQ. The criteria for high and low are whether the score is 1 SD higher or lower than the average, following the recommendation by Aiken et al. [70]. CQ is high regardless of the length abroad when the dietary balance is high, but CQ is lower in the group with a low length of studying abroad than in the group with a high length of studying abroad when the dietary balance is low. This indicates that people with less overseas experience are more likely to increase their CQ by improving their dietary balance.

## 7. Discussion

A growing number of researchers are focusing on CQ, which is the ability to adjust and adapt successfully in international settings. Beyond understanding the outcomes of CQ in different domains, recent research has begun exploring factors that might influence CQ. Some factors that have been identified in the existing literature have been classified into two categories: (a) intercultural experiences [8,9,10] and (b) personality traits [11,12,13]. However, it has also been indicated that these two factors may not be sufficient to explain the variance in CQ [3,5]. Regarding CQ closely related to the activities of the brain [14,15], elements of lifestyle could be considered as another possible antecedent of CQ because research in brain science has clarified the relationship between lifestyles and brain conditions [16,17,18], which indicates the consequent association between lifestyles and CQ. This research tests the association of CQ with lifestyle-related variables using businesspersons in Japan, a country that has so far been culturally rather homogenous but is gradually diversifying with the introduction of foreign labor forces in recent years.

The results of path analyses indicate that dietary balance and walking frequency are significantly associated with the latent variable CQ after controlling for personality variables (curiosity and grit) and the length of studying/working abroad. These lifestyle-related variables are items that have been shown to correlate with brain health [26,27,28,29,30,31] and intelligence [32,33,34] and are consistent with the argument that CQ factors can be grasped by brain science [3,14,15]. In addition, the variables representing curiosity, grit, length of studying abroad, and length of working abroad that were used as covariates were all correlated with CQ, consistent with previous research. These results demonstrate the effectiveness of demonstrating the robustness of an analytical model that emphasizes traditional personalities [20,22,23] and international experiences [55,56,57,58,59] while adding new lifestyle variables. Furthermore, the moderation test showed that the effect of dietary balance on increasing CQ is greater for people with less overseas experience. This suggests that dietary balance compensates for lack of overseas experience in the effect of maintaining the level of CQ, which is difficult to explain without assuming a complicated brain mechanism, as was proposed by CQ developers [14,15].

Although not the main theme, the length of working abroad was not as strongly correlated with CQ subsets as was the length of studying abroad (see Table 2), and, unlike the length of studying abroad, it did not moderate the relationship between dietary balance and CQ (see Figure 2 and Figure 3). This result indicates that studying abroad is more effective in increasing CQ than working abroad and is consistent with Moon et al. [10], who concluded that non-work-related international experiences predict CQ better than work-related experiences, based on the analysis using a pre- and post-test design. Moon et al. [10] attributed the difference to the theory of resource allocation [71,72]. From a resource allocation perspective, it is expected that, when individuals visit foreign countries for work-related purposes, they would not have enough time to interact with people in the country because their primary purpose for being in the country is strongly tied to their work-related activities, such as gathering information on products and monitoring of the foreign subsidiaries [10]. These results did not change even when the length of overseas experience was replaced by the presence or absence of overseas experience or when data on active overseas expatriates were excluded, consistent with previous neuroscience studies, which showed that even temporary, incidental exposure to another culture can change the brain activation pattern when performing the same cognitive task [73,74].

## 8. Implications for Theory and Practice

This research contributes to the existing literature in three ways. First, the study showed that lifestyle indicators like dietary balance and walking frequency are associated with CQ. This is because CQ is related to brain function [15] and intelligence [32,33,34], which are associated with a good combination of exercise and food [26,27,28,29,30,31]. The results of the study indicate a possibility that the mechanism of CQ could be analyzed neurologically in the future, as proposed by the CQ scale developer [3,14,15]. In parallel with this, it is expected that educational methods to enhance cross-cultural understanding will evolve in not just personality-related or international experience-related ways (as currently), but also in ways that encourage a lifestyle that supports a healthy body and brain.

Second, we replicated the correlation between overseas experience, curiosity, and grit, which were often used in previous studies, with CQ. Previous studies have shown a significant but not strong correlation between these variables and CQ [3,7], suggesting the existence of other explanatory variables. The current study showed that one of them is a healthy life such as eating and exercising, from the perspective of brain science. However, the attempts to clarify CQ factors may be made from a more comprehensive perspective in future studies.

Third, we analyzed the factors of CQ for the first time using Japanese samples. The Japanese have long been accustomed to a single culture. However, in recent years of globalization, the introduction of foreign workers and the expansion of companies overseas have increased opportunities for contact with different cultures. Today, it is pointed out that ethnocentricity makes it difficult for Japanese people to manage different cultures [37,38], and the results of this paper give a hint to improve their CQ while incorporating biological perspectives in addition to traditional methods.

## 9. Limitations of the Study and Directions for Future Research

The current study has eight limitations. First, as Japanese businesspersons were used as the subjects, the obtained results may not apply to other societies. Second, the number of participants is relatively small and most of them are male, which restricts the generalizability of the current findings. Third, as we used mostly the sample of domestic dwellers, different results may be obtained if people from multicultural environments such as those living in foreign countries are used. Fourth, as this is a cross-sectional study, the results may not be enough to show causality between determinants and outcomes [75]. Fifth, as this study uses self-reported data from individuals, common method bias between variables cannot be neglected, although the results of confirmatory factor analyses indicate that the problem is not significant. Sixth, as the heterogeneity (e.g., the large age range) in our sample may have affected the results, the results of the current research should be verified after narrowing down the scope in future research. Seventh, there is a possibility of spurious relationships between dietary balance/exercise and CQ. For instance, unobserved third variables (such as socioeconomic status) could be associated with both dietary balance/exercise and CQ. Finally, there are other variables that represent lifestyle. In particular, the importance of the relationship between sleep and both diet and exercise, as well as its influence on the brain and intelligence, has been emphasized in recent years [76], suggesting that there is still room for further improvement of the analytical model in this study. To sum up, future research should verify the robustness of the current research by targeting more diverse or, conversely, more limited subject backgrounds, incorporating objective, socio-economic, and other lifestyle information and longitudinal analysis methods with a larger sample size.

## 10. Conclusions

A growing number of researchers are focusing on CQ, which is the ability to adjust and adapt successfully to different cultural contexts, because CQ is a vital quality for a person in the diversified society that is seen nowadays. However, compared to the effects of CQ (intercultural adaptation, etc.), there are few analyses of factors that enhance CQ. Furthermore, a series of previous studies have shown that the correlation between CQ and overseas experience and personalities, which are often used as antecedents of CQ, is not high enough. If CQ can be treated health-scientifically as, the CQ developers indicated, variables that are good for the brain can also be considered to correlate with CQ. Therefore, in this paper, in addition to overseas experience and personalities, we searched for antecedents of CQ by using dietary balance and walking frequency as independent variables, which have been shown to correlate with brain health in previous studies. Using 142 businesspersons in Japan, the results of the structural equation modeling indicate that, in addition to curiosity and grit, dietary balance and walking frequency showed a significant correlation with CQ after controlling for international experiences. In addition, it was also shown that there are individual relationships between these variables and CQ subsets and that the dietary balance and overseas experience complement each other in terms of their impact on CQ. These indicate the possibility of dealing with CQ from a health-scientific point of view.

## Figures and Tables

**Figure 1 behavsci-14-00028-f001:**
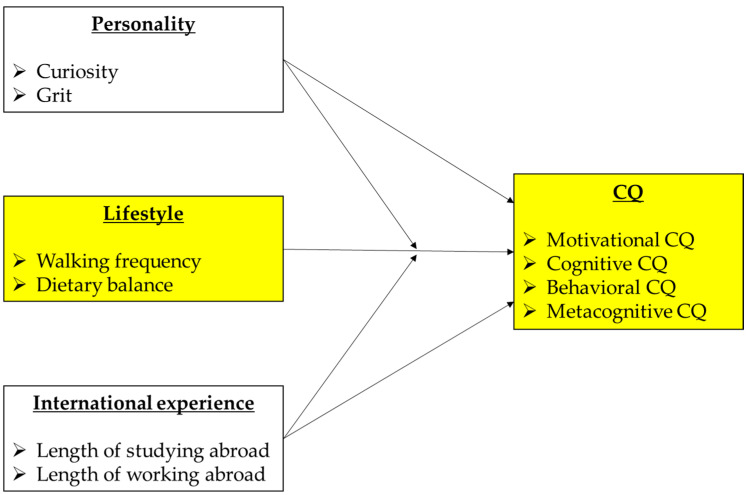
Analytical model of this study. Yellow variables are main variables in this study, others are covariates; circles are latent variables, and squares are observed variables.

**Figure 2 behavsci-14-00028-f002:**
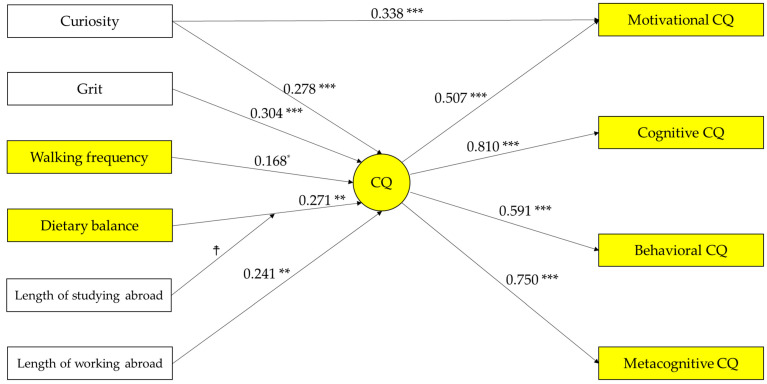
The results of the structural equation modeling. Note: *n* = 142, *** *p* < 0.001, ** *p* < 0.01, * *p* < 0.05. Goodness-of-fit indices: χ^2^ = 29.923; df = 33; root-mean-square error of approximation (RMSEA) = 0.000; probability of close fit (PCLOSE) = 0.932; goodness of fit index (GFI) = 0.965; adjusted goodness of fit index (AGFI) = 0.929; normed fit index (NFI) = 0.948; comparative fit index (CFI) = 1.000. Errors and covariances are omitted in the figure. The covariances are as follows: curiosity and walking frequency (0.185 *), length of working abroad and grit (0.177 *), walking frequency and length of working abroad (−0.178 *), curiosity and grit (0.168 *), length of studying abroad and interaction term (0.856 ***), dietary balance and interaction term (−0.185 ***), error of motivational CQ and error of behavioral CQ (0.441 ***), walking frequency and error of behavioral CQ (0.180 *), error of metacognitive CQ and length of working abroad (−0.248 *), and error of cognitive CQ and dietary balance (−0.283 *). ☨ path from interaction term to CQ (−0.345 *); path from length of studying abroad to CQ (0.519 **). Yellow variables are main variables in this study, others are covariates; circles are latent variables, and squares are observed variables.

**Figure 3 behavsci-14-00028-f003:**
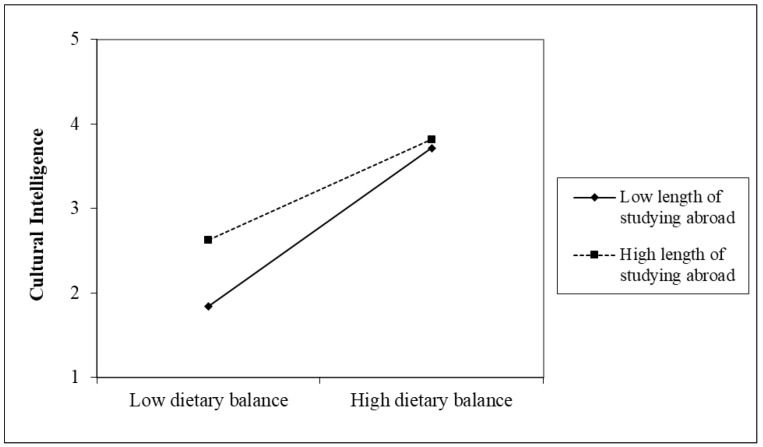
The moderating effect of length of studying abroad between dietary balance and cultural intelligence.

**Table 1 behavsci-14-00028-t001:** Results of confirmatory analyses.

Model	χ^2^:df Ratio	CFI	RMSEA	SRMR
6-factor model	0.919	1.000	0.000	0.070
3-factor model	1.240	0.964	0.041	0.071
1-factor model	1.614	0.910	0.065	0.093

Note: 6-factor model (1 = metacognitive CQ, 2 = cognitive CQ, 3 = motivational CQ, 4 = behavioral CQ, 5 = curiosity, 6 = grit); 3-factor model (1 = CQ, 2 = curiosity, 3 = grit); 1-factor model (1 = all factors combined).

**Table 2 behavsci-14-00028-t002:** Descriptive statistics and correlations.

		Mean	SD	1	2	3	4	5	6	7	8	9	10	11	12
1	Gender	0.782	0.415												
2	Age	42.590	10.696	0.200 *											
3	Length of working abroad	7.300	26.330	0.002	0.193 *										
4	Length of studying abroad	2.122	8.697	−0.143	−0.043	0.126									
5	Curiosity	32.490	8.001	0.082	−0.100	−0.004	0.080	(0.902)							
6	Grit	25.740	5.539	−0.059	0.128	0.170 *	0.161	0.176 *	(0.787)						
7	Dietary balance	10.130	1.304	0.104	0.222 **	0.059	0.084	0.069	0.198 *						
8	Walking frequency	4.700	2.393	0.113	−0.046	−0.198 *	0.030	0.203 *	0.035	−0.072					
9	Metacognitive CQ	18.990	5.108	0.070	0.086	0.029	0.221 **	0.314 ***	0.365 ***	0.324 ***	0.200 *	(0.903)			
10	Cognitive CQ	22.810	8.965	0.036	0.013	0.245 **	0.280 **	0.291 ***	0.355 ***	0.176 *	0.085	0.552 **	(0.931)		
11	Motivational CQ	22.610	6.838	−0.093	−0.008	0.104	0.190 *	0.551 ***	0.359 ***	0.189 *	0.173 *	0.467 **	0.526 **	(0.922)	
12	Behavioral CQ	22.790	5.659	−0.138	0.010	0.128	0.236 **	0.289 ***	0.297 ***	0.199 *	0.258 **	0.498 **	0.474 **	0.661 **	(0.862)

Note: *n* = 142; reliabilities are shown along the diagonal in parentheses. *** *p* < 0.001, ** *p* < 0.01, * *p* < 0.05.

## Data Availability

The datasets used and/or analyzed in the current study are available from the corresponding author upon reasonable request.

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
