# Peer review of "The Association between Lifestyles (Walking/Diet) and Cultural Intelligence: A New Attempt to Apply Health Science to Cross-Cultural Research"

_behavsci, 2023, doi:10.3390/bs14010028_

Round 1
Reviewer 1 Report
Comments and Suggestions for Authors
This paper has a significant contribution, especially to cultural intelligence. However, this paper has several shortcomings that require revision.
1. The author(s) need to improve the abstract on several points, including briefly stating the research methodology, explaining the research design and approach used in this paper, and drawing a conclusion.
2. The introduction should briefly define the purpose of the work and its significance.
3. The author(s) mention Dietary Balance and Walking in the title, but in the literature review, it is only described. The author(s) should add more literature on dietary balance and walking.
4. The author(s) should add the figure of the research framework to give clear information.
5. The discussion acknowledges consistency with previous studies regarding the associations of curiosity and grit with CQ. However, it lacks a comprehensive comparison or contrast with the existing literature on the relationship between lifestyle variables and CQ.
Reviewer 2 Report
Comments and Suggestions for Authors
Overall, authors did an excellent job on this interesting research topic. There are several minor comments:
1. Consider revise the title so that independent variables are ahead of the dependent outcome. E.g.: the association between lifestyles (walking / diet) and CQ.
2. In the literature review section, it would be great if authors elaborate further on why selecting walking/dietary balance in particular out of all the other factors.
3. In terms of hypotheses and research question: it was not clear to readers why H1/H2 would be related to the overall hypothesis on the association between lifestyle factors and CQ. Consider removing them.
4. Were the three affiliates in China excluded? They might have different exposure to dietary resources. If not, how were potential confounders such as living environment, age, neighborhood etc. controlled for in this study?
5. Authors did an excellent job on results section and limitations.
Comments on the Quality of English LanguageLine 289: Consider revising the term input.
